# Antimicrobial Properties of *Apis mellifera*’s Bee Venom

**DOI:** 10.3390/toxins12070451

**Published:** 2020-07-11

**Authors:** Hesham El-Seedi, Aida Abd El-Wahed, Nermeen Yosri, Syed Ghulam Musharraf, Lei Chen, Moustafa Moustafa, Xiaobo Zou, Saleh Al-Mousawi, Zhiming Guo, Alfi Khatib, Shaden Khalifa

**Affiliations:** 1International Research Center for Food Nutrition and Safety, Jiangsu University, Zhenjiang 212013, China; 2Department of Molecular Biosciences, The Wenner-Gren Institute, Stockholm University, S-106 91 Stockholm, Sweden; 3Al-Rayan Research and Innovation Center, Al-Rayan Colleges, Medina 42541, Saudi Arabia; 4Department of Chemistry, Faculty of Science, Menoufia University, Shebin El-Kom 32512, Egypt; aidaabd.elwahed@arc.sci.eg (A.A.E.-W.); nermeen.yosri@science.menofia.edu.eg (N.Y.); 5Department of Bee Research, Plant Protection Research Institute, Agricultural Research Centre, Giza 12627, Egypt; 6School of Food and Biological Engineering, Jiangsu University, Zhenjiang 212013, China; zou_xiaobo@ujs.edu.cn (X.Z.); guozhiming@ujs.edu.cn (Z.G.); 7H.E.J. Research Institute of Chemistry, International Center for Chemical and Biological Sciences, University of Karachi, Karachi 75270, Pakistan; musharraf@iccs.edu; 8College of Food Science, Fujian Agriculture and Forestry University, Fuzhou 350002, China; leichen@fafu.edu.cn; 9Department of Chemistry, Faculty of Science, University of Kuwait, Safat 13060, Kuwait; mostafa_msm@hotmail.com (M.M.); salehalmousawi@hotmail.com (S.A.-M.); 10Department of Pharmaceutical Chemistry, Faculty of Pharmacy, International Islamic University Malaysia, Kuantan, Pahang 25200, Malaysia; alfikhatib@iium.edu.my; 11Faculty of Pharmacy, Airlangga University, Surabaya 60155, Indonesia

**Keywords:** bee venom, antimicrobial properties, melittin, apamin, phospholipase A2 (PLA2)

## Abstract

Bee venom (BV) is a rich source of secondary metabolites from honeybees (*Apis mellifera* L.). It contains a variety of bioactive ingredients including peptides, proteins, enzymes, and volatile metabolites. The compounds contribute to the venom’s observed biological functions as per its anti-inflammatory and anticancer effects. The antimicrobial action of BV has been shown in vitro and in vivo experiments against bacteria, viruses, and fungi. The synergistic therapeutic interactions of BV with antibiotics has been reported. The synergistic effect contributes to a decrease in the loading and maintenance dosage, a decrease in the side effects of chemotherapy, and a decrease in drug resistance. To our knowledge, there have been no reviews on the impact of BV and its antimicrobial constituents thus far. The purpose of this review is to address the antimicrobial properties of BV and its compounds.

## 1. Introduction

According to the World Health Organization (WHO), the antimicrobial drug resistance of bacterial pathogens has reached alarming rates in several parts of the world, and few alternatives are available [1]. The discovery of antibiotics served as a promise to eliminate numerous ailments that menaced human life in the past. However, unexpected side effects such as resistance and mutation displayed a new challenge for humankind. The annual deaths attributable to anti-microbial resistance are expected to surpass those of cancer by 2050 [2]. Due to the resulting overuse of antibiotics, microbes have become capable of developing biofilms embedded in an extracellular matrix (ECM) that are more resistant and more difficult to penetrate with antibiotics. The rise of antimicrobial drug resistance calls for a search of new candidates with novel mode of action. Natural products including bee venom (BV), one of many bee products which is rich in bioactive compounds, offer a diversity of activities against variety of diseases causes [3,4,5,6].

Venoms and their peptides from different animals or organisms such as bees, snakes, wasps, and scorpions, represent promising antimicrobial agents against various microbial pathogenesis [7,8,9,10,11,12]. BV is the venomous cocktail secreted by honeybee workers’ poison glands as a protection mechanism [13]. BV is injected into the victim’s skin using stingers, which ultimately leads to the death of the bee itself afterwards. Although BV is toxic to predators, it has acquired medicinal benefits over the years [14]. Therapeutic usage of BV dates back to Ancient Egypt (4000 BC), and was later applied by Hippocrates, Aristotle, and Galen, during the Greek and Roman historical periods [15]. In Traditional Chinese Medicine and other historical practices, BV was introduced for inflammatory diseases such as rheumatoid, arthritis, tendonitis, fibrosis, lupus, and multiple sclerosis [16,17].

It is thought that medical use and subsequent public acceptance of BV is due to the availability of biologically active compounds such as peptides. For instance, melittin is a major compound representing 40–60% of the dry BV weight [18]; it also contains mast cell degranulating peptide (MCDP), secapin, and its isomers (i.e., secapin-1 and -2), adolpanin, tertiapin, apamin [19,20,21,22,23,24,25], and enzymes, i.e., phospholipase A2 (PLA2), hyaluronidases, and acid phosphatase [26,27,28]. Furthermore, dipeptidylpeptidase IV (Api m 5) [29], Api m 6 [30], CUB serine protease (Api m 7) [31], icarapin (Api m 10) [32], major royal jelly proteins (MRJP 8 and 9) [33,34], and volatile compounds (isopentyl acetate and (Z)-I l-eicosen-l-ol) [35] are also present. Attributing to these constituents, BV has been proven to be active as an anti-inflammatory [36,37], radioprotective, [38], and antibacterial agent against several Gram-positive/negative bacteria strains [5,39]. The combination of BV and its constituents with chemotherapy agents (vancomycin, oxacillin, and amikacin) has a synergetic effect against bacteria due to the antibacterial properties [16].

In this review, we discuss the in vitro, in vivo, and in situ therapeutic implications of BV against microbial diseases.

## 2. Antimicrobial Properties of Bee Venom and Mode of Action for the Venom and its Derived Compounds

### 2.1. Antibacterial

BV has significant antimicrobial effects [40]. BV, and its major compounds, PLA2 and melittin (Figure 1), were applied against oral pathogens identified as the causative agents of tooth decay. The minimum inhibitory concentration (MIC) for the BV lies between 20 and 40 µg/mL against *Streptococcus salivarius*, *S. sobrinus*, *S. mutans*, *S. mitis*, *S. sanguinis*, *Lactobacillus casei*, and *Enterococcus faecalis*. Melittin showed MIC values ranging from 4 to 40 µg/mL, whereas the MIC value of PLA2 was above 400 µg/mL (Table 1) [4]. Lyme infection is a tick-borne multi-systemic illness caused by the bacterium *Borrelia burgdorferi* [41]. Both BV and melittin had impacts on the morphology and size of the biofilms of *B. burgdorferi*, whereas antibiotics frequently experienced backslide occurrence after discontinuation [6].

BV antimicrobial and antibiofilm activity was identified in 16 poultry-isolated Salmonella strains. BV MIC ranged 256–1024 μg/mL. Sub-inhibitory BV concentrations significantly reduced the development of biofilm in 14 of the 16 Salmonella strains studied, with substantial motility increases. BV did not show any influence on the motility of *Salmonella isangi* IG1 and *S. infantis* Lhica I17. The percentage of biofilm reduction observed ranged from 27.66% (*S. Infantis* Lhica I17) to 68.22% (*Salmonella enterica* subsp. salamae SA3), with significant variability among the different Salmonella strains tested [42].

BV was proven effective, synergistic, and safe when combined with some conventional drugs against certain types of microbes; however, there is a slow and careful consideration towards its investigations in pre-clinical and clinical applications. For example, BV and melittin exhibited a broad-spectrum antibacterial activity against both Gram-positive (MIC values between 10 and 100 μg/mL) and Gram-negative bacteria (MIC values between 30 and *>*500 μg/mL). Combination of BV and melittin with other antibiotic drugs, i.e., oxacillin, vancomycin, and amikacin, using checkerboard dilution gave fractional inhibitory concentration (FIC) indices ranging between 0.24 and 0.5; the FIC index is determined by the MIC of the test material in combination with an antibiotic medication divided by the MIC of the test material individually [16]. BV increased the antibody production against formalin-killed *S. gallinarum* in broiler chicks [43]. BV and melittin exhibited a broad spectrum antibacterial activity against Methicillin-resistant *Staphylococcus aureus* (MRSA) and vancomycin-resistant enterococci at MIC values of 6–800 μg/mL, compared to vancomycin’s (reference drug) MIC value of 1.6–25 μg/mL [16]. The combination of BV and melittin with oxacillin showed a bactericidal effect on MRSA ATCC 33591. The treatment with both BV and melittin led to changes in the bacterial cell membrane caused by the loss of membrane integrity and exhibition of changes in the cell morphology including cell distortion and loss of cytoplasm content [11].

The antibacterial efficiency of melittin was studied against a variety of bacteria, such as *Escherichia coli*, *S. aureus*, and *B. burgdorferi* [39,44,45,46,47]. Gram-positive bacteria have sensitivity to melittin, compared to Gram-negative ones, due to the nature of the organism’s cell membrane [39,46,48,49]. Melittin can penetrate the peptideoglycan layer of the Gram-positive cell membrane more easily than the Gram-negative cells, which have a layer of lipopolysaccharides protecting their membrane. The presence of proline residue in position 14 has been shown to play a central role in the antimicrobial activity of melittin. Its absence in a melittin analog significantly reduced antimicrobial activity compared to the native peptide [50]. Similarly, two synthetic melittin, serine-substituted melittin (Mel-S) and asparagine-substituted melittin (Mel-N), were capable of penetrating *E. coli* cell membrane. Mel-S was more efficient than Mel-N [51]. Melittin, in general, possesses a greater ability to destroy biofilms formed by *S. aureus* compared to *E. coli* (biofilm production was 56% vs. 37%, respectively) [46]. MDP1: GIGAVLKVLTTGLPALIKRKRQQ and MDP2: GIGAVLKWLPALIKRKRQQ displayed strong antibacterial activity against reference strains of *S. aureus*, *E. coli*, and *Pseudomonas aeruginosa* compared to the native melittin. The antibacterial effects of MDP1 and MDP2 were explained by the changes in the bacterial membrane and the destruction of the bacterial cell membrane. Furthermore, the hemolytic activity of melittin (93.5%) at the dose of 3.84 µg/mL with average MIC values showed significant reduction in MDP1 (1.46% at geometric mean (GM) of 3.01 µg/mL) and MDP2 (5.15% at GM of MICs 2.18 µg/mL) [52]. The antibacterial activity of native melittin and its two mutants, namely melittin I17K (GIGAVLKVLTTGLPALKSWIKRKRQQ) with a higher charge and lower hydrophobicity and mutant G1I (IIGAVLKVLTTGLPALISWIKRKRQQ) of higher hydrophobicity, were investigated against different strains of Listeria, as mentioned in Table 1 [53].

The increased frequency of multi-drug resistant (MDR) bacteria is a major challenge to antimicrobial treatment. Melittin shows broad antibacterial activity toward different types of bacteria such as methicillin-susceptible *S. aureus* (MSSA), MRSA, and *Enterococcus* spp at MICs 0.5–4, 0.5–4, and 1–8 µg/mL, respectively. Furthermore, synergetic action between melittin and some antibiotics, i.e., daptomycin, vancomycin, linezolid, ampicillin, and erythromycin, against the previously mentioned bacteria were investigated by Dosler et al. [54]. Melittin‘s antibacterial and synergistic effects with *β*-lactam antibiotics to *Acinetobacter baumannii* was reported using broth microdilution method. The MIC values of melittin, ciprofloxacin, co-amoxiclav, imipenem, netilmicin, ceftazidime, and piperacillin are 4, 8, 16, 16, 16, 32, and 128 µg/mL, respectively. However, FIC indices for combinations of melittin with the same antibacterial drugs are 0.750, 0.312, 0.250, 1.25, 0.187, and 0.375 µg/mL, respectively [55]. The application of melittin–doripenem has resulted in a significant decrease in the MIC of MDR baumannii strains. When the combinations of melittin–doripenem and melittin–ceftazidime were administrated to strains of MDR *P. aeruginosa*, the dose of melittin was significantly reduced. The combination of melittin with doripenem and ceftazidime against MDR microbial pathogens could be of great therapeutic value [56].

Furthermore, the combination between melittin and PLA2 (0.5 mg of each compound) has been investigated against oral pathogens *S. salivarius*, *S. sobrinus*, *S. mutans*, *S. mitis*, *S. sanguinis*, *Lactobacillus casei*, and *E. faecalis.* The MIC was studied for each one individually (melittin with MIC from 4 to 40 µg /mL and PLA2 with MIC values of >400 µg/mL) and in combination with each other (MIC values ranging 6–80 µg/mL) [4]. The combination of BV with ampicillin or penicillin yielded an index of inhibitory concentrations ranging from 0.631 to 1.002, indicating a partial synergistic effect. The two MRSA strains were more susceptible to the combination of BV with gentamicin or vancomycin compared to combination of BV with ampicillin or penicillin [5].

### 2.2. Anti-Viral

During the last decade, viral diseases such as hepatitis C, smallpox, polio, rubella, and AIDS have threatened the lives of millions worldwide, especially immunocompromised patients [71]. Water contamination (waterborne diseases) represents a major health problem in regards to the spread of many viral diseases like hepatitis viral disease, poliomyelitis, gastroenteritis, diarrhea, etc. [72,73,74]. Searching for anti-viral substitutes that are low or completely free of diverse effects is an urgent need. In this context, natural products, in particular BV, embody a variable of exotic constituents, suggesting an immeasurable source of anti-viral agents [75].

BV and its constituents show prominent anti-viral activities against various enveloped and non-enveloped viruses such as Vesicular Stomatitis (VSV), Herpes Simplex (HSV), Enterovirus-71 (EV-71), Coxsackie (H3), Respiratory Syncytial Influenza A (A/PuertoRico/8/34) (in vitro study), and influenza A subtype (H1N1) (in vivo study) (Table 2 and Figure 2) [76]. Papillomaviruses (HPVs) are considered the most common agents responsible for cervical carcinoma. BV was able to inhibit the growth of cervical cancer cells by the downregulation of E6/E7 proteins of HPV viruses (Table 2) [77]. BV and its constituent melittin (Figure 1) can induce the immunity against porcine reproductive and respiratory syndrome viruses (PRRSV) via significant up-regulation of Th1 cytokines (IFN-γ and IL-12) and several types of immune cells, including CD3^+^CD8^+^, CD4^+^CD8^+^, and γδ T cells, leading to reduction of the viral load and decrease of the severity of interstitial pneumonia in PRRSV-infected pigs [78].

Based on the HIV tropism, honeybee PLA2 and its derivatives p3bv (containing 21–35 amino acids of PLA2) possess potent anti-human immunodeficiency virus (HIV) activity. The p3bv peptide showed anti-HIV activity via the prevention of the cell–cell fusion process and inhibition of the replication of T-tropic viruses in contrast to PLA2 that inhibited both M- and T-tropic HIV viruses but was unable to inhibit cell–cell fusion under the same condition. The authors illustrated that the mechanism behind the inhibition of HIV replication is different for the two enzymes. PLA2 is presumably linked to a high-affinity binding receptor of the host cells but P3bv peptide is linked with a CXCR4 chemokine receptor [79,80]. Another type of BV phospholipases A2 called sPLA2 was evaluated using plaque assay and proven to suppress the activity of Japanese encephalitis virus (JEV), Hepatitis C virus (HCV), and Dengue virus (DENV) with IC_50_ values of 49, 117, and 183 ng/mL, respectively [81].

Furthermore, melittin represents an agent against *Arenavirus Junin* (JV) and Herpes Simplex Types 1 (HSV-1) and 2 (HSV-2) via inhibition of virus multiplication, adsorption, and penetration, as well as Na^+^ and K^+^ pumps of the host cell. Utilizing plaque and viral penetration assays, melittin at a multiplicity of infections (m.i.o) of 0.02 and 0.05 µM inhibited plaque formation giving 37 plaque formation units (PFU) compared to 220 PFU observed in the absence of melittin [82,83]. In another in vitro study, melittin was evaluated using the plaque assay against different viruses, namely Respiratory syncytial virus (RSV), EV-71, HSV, H3, Fused Influenza A virus (PR8), and VSV, with EC_50_ values of 0.35 ± 0.08, 0.76 ± 0.03, 0.94 ± 0.07, 0.99 ± 0.09, 1.15 ± 0.09, and 1.18 ± 0.09 μg/mL, respectively [76]. Furthermore, melittin could also be used against HIV-1, as it can inhibit the replication of HIV-1 by interfering with host cell-directed viral gene expression [84]. In an in vivo study, melittin was examined against lethal doses of the pathogenic H1N1 virus in mice; the results show that melittin can inhibit the replication of the virus, as the Log_10_ 50% tissue culture infectious dose of a virus (TCID_50_) was 1.53 ± 0.25, compared to phosphate-buffered saline (PBS) at Log_10_ TCID_50_ 4.22 ± 0.2 [76]. Taken together, these results suggest that BV and its constituents have the potential to become therapeutic agents to combat infectious viral diseases.

### 2.3. Anti-fungal

Fungal related diseases cause colonization, superficial skin infections, and allergies, representing a devastating health problem worldwide. Additionally, the toxicity and resistance to antifungal drugs are major challenges. Natural products from plants, marine life, microorganisms, and bee products could be considered promising antifungal agents with fewer side effects [88].

Recently, BV was reported as an effective agent against many of the fungal related diseases, as mentioned in Table 3. BV can inhibit dermatophytosis, which occurs via *Trichophyton mentagrophytes* and *Trichophyton rubrum* fungi. BV reduced all populations of *T. mentagrophytes* at 15 and 30 ppm within 5 min, while, at the same dose of BV, *T. rubrum* growth inhibition was observed within 5 min. On the other hand, fluconazole did not prevent the development of the same pathogens. The study proved that the BV was more potent than fluconazole (commercial antifungal drugs) [89]. The anti-fungal action of BV on 10 clinical isolates of *Candida albicans* was studied, with MIC values ranging from 62.5 to 125 μg/mL [88]. In another study, melittin showed antimicrobial activity against various strains of fungi with MIC values between 30 and 300 μg/mL [16]. Melittin produced oxygen free radicals (OH)**^.^** that could induce apoptosis of *C. albicans*. The fungal cell death was explained by the disrupted mitochondrial membrane via the Ca^2+^ release [90,91].

*Alternaria alternate* sp. and *Aspergillus pillows* are common pathogens that grow in the nasal cavity. The irritation/inflammation caused by the fungus induces the production of chemical mediators from nasal epithelial cells and fibroblasts. Melittin and apamin (Figure 1) were able to inhibit the growth of *A. alternate sp.* and *A. pillows* causing upper airway inflammatory diseases. The mechanism of action was shown to be via the inhibition of chemical mediators production, i.e., interleukin (IL)-6, IL-8, and ECM, as well as induction of the phosphorylation of Smad 2/3 and p38 MAPK [3].

## 3. Concluding Remarks

Microbial diseases are problematic, particularly with the emergence of drug resistance; therefore, researchers are looking for new sources of bioactive candidates. Natural products are considered a renewable source with fewer complications that could provide a wide number of active compounds. BV is a complex mixture of proteins, peptides, and low molecular weight components including melittin, PLA2, apamin, adolapin, and MCDP. BV contains a variety of bioactive components including melittin, apamin, and PLA2, which play a vital role as antimicrobials through various mechanisms against bacteria, viruses, and fungi. The synergistic effect of BV and melittin through the combination of chemotherapy drugs leads to a reduction in dosage, side effects, and greater efficacy of the treatment strategy against microbial ailments. BV established its role as antimicrobial.

BV and its constituents in combination with antibiotic drugs emerge as a plausible approach to overcome drug resistance of current antibiotics treatment in a controlled manner. Another promising and feasible implication is to test BV to combat microbes causing skin diseases. Interestingly, BV can be useful as a topical agent for encouraging skin regeneration or treatment of certain epidermal conditions [5,94]. Therefore, BV has contributed to some formulations against bacteria that cause acne [95,96].

Therefore, BV and melittin are attractive therapeutic candidates for microbial diseases. However, using BV and melittin induces extensive hemolysis and toxicity of the cells, a severe side effect that limits their future development and clinical application. Ongoing research is addressing practical issues including standardization, toxicity, and stability [97,98,99,100].

## Figures and Tables

**Figure 1 toxins-12-00451-f001:**
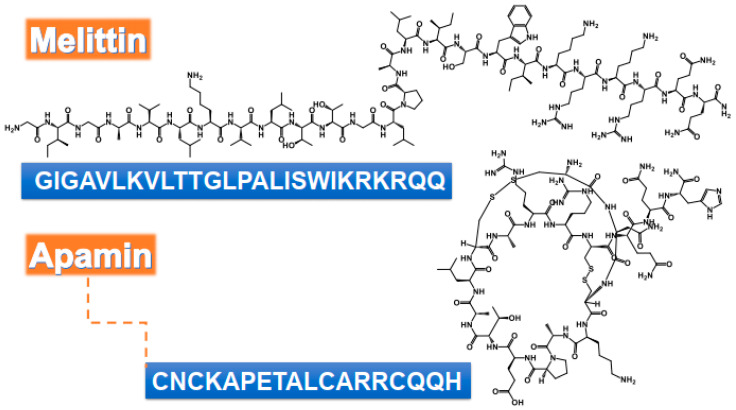
Chemical structure and amino acids sequence of bioactive peptides from bee venom as antimicrobial agents.

**Figure 2 toxins-12-00451-f002:**
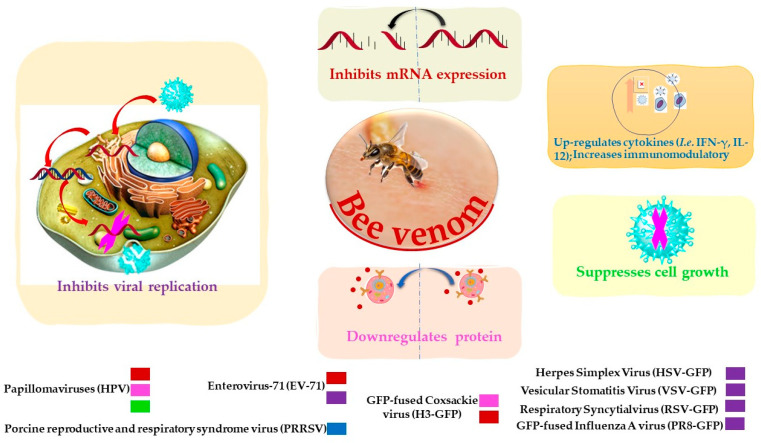
Possible inhibitory mechanisms of bee venom against a variety of viruses.

**Table 1 toxins-12-00451-t001:** Anti-bacterial properties of bee venom and its compounds.

Bee Venom/Isolated Compounds	Organism	Method	Dose/Mode of Action	Reference
Bee venom	*S. aureus*	Disc diffusion	MIC 8 µg/mL	[57]
			MBC 16 µg/mL	
	*S. aureus Gp*	Disc-diffusion	At 100 µg/mL give inhibition zone 23.2 mm after 24 h	[10]
	MRSA CCARM 3366	Broth microdilution	MIC 0.085 μg/mL	[5].
			MBC 0.106 μg/ mL	
	*S. aureus* CCARM 3708	Broth microdilution	MIC 0.11 μg/mL	[5].
			MBC 0.14 μg/mL	
	MR *S. aureus* ATCC 33591	Broth microdilution	MIC_90%_ 7.2 μg/mL	[11]
			MBC_90%_ 28.7 μg/mL	
			PC: Cephalothin	
			MIC_90%_ 2 μg/mL	
			MBC_90%_ 2 μg/mL	
	*S. aureus* enterotoxin ATCC 23235	Broth microdilution	MIC 0.7 μg/mL	[11]
			PC: Cephalothin and Oxacillin	
			MIC < 0.5 μg /mL	
	*S. hyicus*	Disc diffusion	MIC 128 µg/mL	[57]
			MBC 128 µg/mL	
	*S. chromogenes*	Disc diffusion	MIC 128 µg/mL	[57]
			MBC 128 µg/mL	
	*S. salivarius*	Broth microdilution	MIC 20 µg/mL	[4]
			PC: Chlorhexidine digluconate	
			MIC 0.9 µg/mL	
	*S. sanguinis*	Broth microdilution	MIC 30 µg/mL	[4]
			PC: Chlorhexidine digluconate	
			MIC 3.7 µg/mL	
	*S. sobrinus*	Broth microdilution	MIC 40 µg/mL	[4]
			PC: Chlorhexidine digluconate	
			MIC 0.9 µg/mL	
	*S. mitis*	Broth microdilution	MIC 40 µg/mL	[4]
			PC: Chlorhexidine digluconate	
			MIC 3.7 µg/mL	
	*S. mutans*	Broth microdilution	MIC 20 µg/mL	[4]
			PC: Chlorhexidine digluconate	
			MIC 0.9 µg/mL	
	*Klebsiella pneumonia*	Broth microdilution	MIC 30 µg/mL for 24 h	[16]
	*Bacillus subtilis*	Broth microdilution	MIC 30 µg/mL for 24 h	[16]
	*Paenibacillus larvae*	Resazurin method	MIC 3.12 μg/mL	[58]
			MBC 4.16 μg/mL	
			PC: Oxytetracycline	
			MIC 0.63 μg/mL	
			MBC 0.83 μg/mL	
	*E. faecalis*	Broth microdilution	MIC 20 µg/mL	[4]
			PC: Chlorhexidine digluconate	
			MIC 3.7 µg/mL	
	*L. casei*	Broth microdilution	MIC 20 µg/mL	[4]
			PC: Chlorhexidine digluconate	
			MIC 0.9 µg/mL	
	*Salmonella* *typhimurium*	Disc-diffusion	Inhibition zone was 15.88 mm at 45 μg	[59]
			PC: Gentamicin	
			Inhibition zone was 19 mm at 10 μg/mL	
	*E. coli*	Disc-diffusion	At 45 μg inhibits 32.46 mm	[59]
			PC: Gentamicin	
			At 10 μg/mL inhibits 20 mm	
	*P. aeruginosa*	NR	The antibacterial activity was 38% at 50 µg/mL	[60]
	*Borrelial spirochetes*	Direct counting method	MIC 200 µg/mL	[6]
			PC: Doxycycline, cefoperazone, and daptomycin	
			MIC 10 µg/mL	
Melittin	*S. salivarius*	Broth microdilution	MIC 10 µg/mL	[4]
	*E. faecalis*	Broth microdilution	MIC 6 µg/mL	[4]
	*L. casei*	Broth microdilution	MIC 4 µg/mL	[4]
	*S. sanguinis*	Broth microdilution	MIC 10 µg/mL	[4]
	*S. sobrinus*	Broth microdilution	MIC 10 µg/mL	[4]
	*S. mitis*	Broth microdilution	MIC 10 µg/mL	[4]
	*S. mutans*	Broth microdilution	MIC 40 µg/mL	[4]
	*K. pneumonia*	Broth microdilution	MIC 8 µg/mL throughout 24 h	[16]
	*B. subtilis*	Broth microdilution	MIC 6 µg/mL for 24 h	[16]
	*Susceptible colistin- A. baumannii*	Broth microdilution	MIC 4 mg/L after 24 h	[61]
	*Acinetobacter* spp.	Disc diffusion	Cell lysis	[62]
			Membranolytic effect	
			MIC 0.5 µg/mL	
	*Colistin-resistant A. baumannii*	Broth microdilution	MIC 2 mg/L after 24 h	[61]
	*Listeria monocytogenes* F4244	Agar well diffusion	MIC 0.315 µg/mL	[53]
			MBC 3.263 µg/mL	
	MR *S. aureus* ATCC 33591	Broth microdilution	MIC_90%_ 6.7 μg/mL	[11]
			MBC_90%_ 26 μg/mL	
			PC: Cephalothin	
			MIC_90%_ 2 μg/mL	
			MBC_90%_ 2 μg/mL	
	*S. aureus* enterotoxin ATCC 23235	Broth microdilution	MIC 3.6 μg/mL	[11]
			PC: Cephalothin and Oxacillin	
			MIC <0.5 μg /mL	
	*S. aureus*	Microtiter broth dilution	MIC 6.25 μg/mL	[63]
	*B. spirochetes*	Direct counting method	MIC 200 µg/mL	[6]
			PC: Doxycycline, cefoperazone, and daptomycin	
			MIC 10 µg/mL	
	*A. baumannii* ATCC 19606	Broth microdilution	MIC 17µg/mL	[64]
			PC: Polymyxin	
			MIC 0.25 µg/mL	
			Imipenem:	
			MIC ≤ 0.125 0.25 µg/mL	
	*A.* *baumannii* 31852 (S)	Broth microdilution	MIC 20 µg/mL	[64]
			PC: Polymyxin	
			MIC 0.25 µg/mL	
			Imipenem:	
			MIC 0.25 µg/mL	
	*A. baumannii* 33677 (XDR)	Broth microdilution	MIC 31 µg/mL	[64]
			PC: Polymyxin	
			MIC 0.25 µg/mL	
			Imipenem:	
			MIC 16 µg/mL	
	*A. baumannii* 96734 (XDR)	Broth microdilution	MIC 45.5 µg/mL	[64]
			PC: Polymyxin	
			MIC 0.25 µg/mL	
			Imipenem:	
			MIC 16 µg/mL	
**Synthetic Melittin and Its Analogues**				
Synthetic melittin	*P. aeruginosa* ATCC 47085	Luria broth	MIC 12.1 µM	[65]
	*E. coli* ATCC 29222	Luria broth	MIC 13.2 µM	[65]
	*E. coli* DH5	NR	MIC 3.9 µM	[66]
			PC: Tetracycline	
			MIC 1.2 µM	
	*K. pneumoniae* ATCC 13883	Luria broth	MIC 14.9 µM	[65]
	*A. baumannii* ATCC 19606	Luria broth	MIC 8.3 µM	[65]
	*B. subtilis*	NR	MIC 2 µM	[66]
			PC: Tetracycline	
			MIC 0.2 µM	
	*S. aureus*	NR	MIC 3.6 µM	[66]
			PC: Tetracycline	
			MIC 4 µM	
Melittin Hybrid				
Cecropin A–melittin (CAM)	*E. coli*	Microtiter broth dilution	MIC 3.7 µg /mL	[67]
CAM-W	*E. coli*	Microtiter broth dilution	MIC 0.3 µg/mL	[67]
Cecropin A-melittin CA(1–8)M(1–18)	*A. baumannii*	Mueller-Hinton broth	MIC 2 µM	[68]
			PC: Polymyxin B	
			MIC 1 µM	
Mutant melittin I17K	*L. monocytogenes* F4244	Agar well diffusion	MIC 0.814 µg/mL	[53]
			MBC 7.412 µg/mL	
Mutant melittin G1I	*L. monocytogenes* F4244	Agar well diffusion	MIC 0.494 µg/mL	[53]
			MBC 5.366 µg/mL	
MM-1	*B. subtilis*	NR	MIC 2.4 µM	[66]
			PC: Tetracycline	
			MIC 0.2 µM	
MM-2	*B. subtilis*	NR	MIC 1.8 µM	[66]
			PC: Tetracycline	
			MIC 0.2 µM	
Mel-H	*E. coli*	Microtiter broth dilution	MIC 11.25 µM	[69]
	*P. aeruginosa* ATCC27853	Microtiter broth dilution	MIC 11.25 µM	[69]
	*S. aureus* ATCC25923	Microtiter broth dilution	MIC 5.6 µM	[69]
Mel(12–24)	*B. subtilis*	Broth microdilution	MIC 0.65 µg/mL	[70]
			PC: Melittin	
			MIC 0.18 µg/mL	
	*S. aureus*	Broth microdilution	MIC 1.3 µg/mL	[70]
			PC: melittin	
			MIC 0.72 µg/mL	
Phospholipase A2	*S. aureus Gp*	Disc-diffusion	Hydrolysis of phospholipids	[10]
			At 100 µg/mL inhibits 13.33 mm after 24 h	
	*L. casei*	Broth microdilution	MIC 400 µg/mL	[4]

PC, Positive control; MIC, Minimum inhibitory concentration; MBC, Minimum bactericidal concentration; NR, No reported; CAM, KWKLFKKIEKVGQGIGAVLKVLTTGL; CAM-W, KWKLWKKIEKWGQGIGAVLKWLTTWL-NH_2_; melittin I17K, GIGAVLKVLTTGLPALKSWIKRKRQQ; CA(1–8)M(1–18), KWKLFKKIGIGAVLKVLTTG LPALIS-NH_2_; Mel(12–24), GLPALISWIKRKR-NH_2_; MM-1, GIGAVLKVLTTGAPALISWIKRKRQQ; MM-2, GIGAVAKVLTTGAPALISWIKRKRQQ; Mel-H, GIGAVLKVLALISWIKRKR.

**Table 2 toxins-12-00451-t002:** Bee venom and its compounds as antiviral agents.

Bee Venom/Isolated Compounds	Organism	Method	Dose/Mode of Action	Reference
Bee venom	Papillomaviruses (HPV16 E6)	Reverse transcription assay	Inhibits mRNA expression.	[77]
			Suppresses cell growth.	
			Downregulates protein.	
			At 10 µg/mL inhibits 0.35 ± 0.06-fold after 24 h.	
	Papillomaviruses (HPV16 E7)	Reverse transcription assay	Inhibits mRNA expression.	[77]
			Suppresses cell growth.	
			Downregulates protein.	
			At 10 µg/mL inhibits 0.44 ± 0.07-fold after 24 h.	
	PRRSV	Enzyme-linked immunosorbent assay	Increases immunomodulatory against the virus.	[78]
			Significant up-regulate Th1 cytokines (IFN-γ and IL-12) and several types of immune cells.	
	Vesicular stomatitis virus (VSV)	Plaque assay	Inhibits virus replication	[76]
			EC_50_ 0.5 ± 0.06 μg/mL	
	HSV	Plaque assay	Inhibits virus replication	[76]
			EC_50_ 1.52 ± 0.11 μg/mL	
	Coxsackie virus (H3)	Plaque assay	Inhibits mRNA expression	[76]
			Inhibits virus replication	
			EC_50_ 0.5 ± 0.04 μg/mL	
	RSV	Plaque assay	Inhibits virus replication	[76]
			EC_50_ 1.17 ± 0.09 μg/mL	
	PR8	Plaque assay	Inhibits virus replication.	[76]
			EC_50_ 1.81 ± 0.08 μg/mL	
	EV-71	Plaque assay	Inhibits mRNA expression.	[76]
			Inhibits virus replication	
			EC_50_ 0.49 ± 0.02 μg/mL	
	Lumpy skin disease virus (LSDV)	Agar gel precipitation test	At the dose 0.5 μg/mL	[85]
Melittin	Immunodeficiency virus (HIV)	Lysis and fusion assays	Lytic and fusogenic	[86]
	*Herpes simplex* (HSV-1)	Plaque assay Virus penetration assay	Inhibits cell fusion.	[83]
			Inhibits Na^+^, K^+^ pump activity.	
			Inhibits virus adsorption and penetration to the cells.	
	Immunodeficiency virus HIV-1	Transient transfection Assays	Inhibits virus replication.	[84]
			Suppresses gene expression.	
			Suppresses intracellular	
			Protein and mRNA synthesis.	
			Suppresses long terminal repeat (LTR) activity	
			ID_50_ 0.9–1.4 µM after 24 h.	
	*Arenavirus Junin* (JV)	Plaque assay	Impedes the multiplication	[82]
			EC_50_ 0.86 µM after 24 h.	
	HSV-1	Plaque assay	Impedes the multiplication	[82]
			EC_50_ 1.35 µM after 24 h.	
	*Herpes simplex virus* (HSV-2)	Plaque assay	Impedes the multiplication	[82]
			EC_50_ 2.05 µM after 24 h.	
	*Herpes simplex virus 1 M* (HSV-1 M)	Quantitative microplate assay	Viral inactivation at 100 µg/mL	[87]
	*Herpes simplex virus 2 G* (HSV-2 G)	Quantitative microplate assay	Viral inactivation at 100 µg/mL	[87]
Phospholipase A2 (sPLA2)	Hepatitis C virus (HCV)	Plaque assay	IC_50_ 117 ± 43 ng/mL after 24 h.	[81]
	DENV	Plaque assay	IC_50_ 183 ± 38 ng/mL after 24 h.	[81]
	JEV	Plaque assay	IC_50_ 49 ± 13 ng/mL after 24 h.	[81]

EC_50_, Effective concentration for 50% reduction; ID_50_, 50% inhibitory dose; IC_50_, Inhibition concentration of 50%.

**Table 3 toxins-12-00451-t003:** Anti-fungi properties of bee venom and its compounds.

Bee Venom/ Isolated Compounds	Organism	Method	Dose/Mode of Action	Reference
Bee venom	*T. mentagrophytes*	Broth dilution	At 0.63 ppm inhibits 92% After 1 h.	[89]
	*T. rubrum*	Broth dilution	At 0.63 ppm inhibits 26% After 1 h.	[89]
	*C. albicans*	Disc diffusion	Prevents dimorphism MIC 40 µg/mL for 48 h.	[92]
	*C. albicans*	Broth microdilution	MIC 62.5–125 μg/mL for 24 h.	[88]
	*Candida krusei*	Broth microdilution	MIC 60 µg/mL throughout 48 h.	[16]
	*A. alternate*	NR	At 1 µg/mL inhibits 50% of interleukin (IL)-6 production.	[3]
			At 1 µg/mL inhibits 28.8% of interleukin (IL)-8 production.	[3]
Melittin	*C. krusei*	Broth microdilution	MIC 30 µg/mL for 48 h	[16]
	*C. albicans*	NR	Disruptive the mitochondrial membrane.	[90]
			Apoptotic for 4 h	
	*Aspergillus flavus* (KCTC 1375)	Microdilution method and MTT assay	MIC 1.25 µM	[93]
			PC: Amphotericin B:	
			MIC 2.5 µM	
			Fluconazole: MIC 10 µM	
			Itraconazole: MIC 10 µM	
	*Malassezia furfur* (KCTC 7744)	Microdilution method and MTT assay	MIC 1.25 µM	[93]
			PC: Amphotericin B: MIC 2.5 µM	
			Fluconazole: MIC 5 µM	
			Itraconazole: MIC 5 µM	
	*C. albicans* (ATCC 90028)	Microdilution method and MTT assay	MIC 2.5 µM	[93]
			PC: Amphotericin B: MIC 5 µM	
			Fluconazole: MIC 10 µM	
			Itraconazole: MIC 10 µM	
Apamin	*A. alternate*	NR	At 1 µg/mL inhibits 42.6% of interleukin (IL)-6 production.	[3]
			At 1 µg/mL inhibits 38.7% of interleukin (IL)-8 production.	[3]

NR, Not reported; ppm, Parts per million.

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
