# Peer review of "Antimicrobial Properties of Apis mellifera’s Bee Venom"

_toxins, 2020, doi:10.3390/toxins12070451_

Round 1

Reviewer 1 Report

The revised version of the manuscript attended most of my concerns and questions and the paper can now be accepted by Toxins. 

Author Response

Response to Reviewer 1 Comments:

The revised version of the manuscript attended most of my concerns and questions and the paper can now be accepted by Toxins.

Response: we would like to thank the referee for kind evaluation and effort as well as compliment.

Reviewer 2 Report

The authors have submitted a revised version of their manuscript, however they did not provide a point-to-point response. The very short response letter I received is inappropriate for the number of comments and essential shortcoming, which have been identified in the reviewers first round of revision. Therefore, this referee has considered the manuscript as full new resubmission.

The referee has identified a huge number of shortcomings in the provided manuscript (for details see adjustment and comments in the provided PDF).

In example:

Paragraph 2. It is not in frame with the abstract and the aims of the review as indicated at the very end of the introduction. The paragraph should be removed from the review. Instead, the authors may introduce the reader to BV and its secondary metabolites more carefully. The BV should be separated in protein, peptide and small molecules. If, possible structures, primary sequence, etc should be shown in a new Figure 1. The first paragraph should discuss the BV constituents, their properties and bioactivities in more general.

Paragraph 3 has to be reorganized. The authors should start with a block of information for BV antimicrobial activities, followed by separate blocks for melittin or other BV derived compounds.

Figure 2 is inappropriate. The figure is informative and can be removed. Overall, the review drastically needs useful figures and the authors are asked to provide such figures for any resubmission.

For any resubmission not the journal, the authors should consider substantial refurbishing of their work. The article lacks a rational structure (remove all parts from BVIT and concentrate on antimicrobial activity). The review is not critical (try to look beyond the curtain) and there is little attempts for an outlook regarding the future of BV,etc . Substantially, the review is a rather boring listening of half-truth. The English language needs extensive editing of language and style. The authors must provide a detailed a point-to-point response and a clean version of the revised manuscript.

This referee recommends major revisions for the submitted manuscript, however the points raised by this reviewer are substantial and correction will need much attention. Any submission of a further inappropriate versions inevitably will lead to rejection. Therefore, the authors may reconsider a resubmission and withdraw their manuscript for publication in the journal of Toxins.

Author Response

Response to Reviewer 2 Comments:

We would like to thank the reviewer for taking the time and effort necessary to provide such insightful guidance. We address each criticism and the comments made by the referee, and explain how we have modified the manuscript to comply with the concerns that were expressed as follow:

  • The referee has identified a huge number of shortcomings in the provided manuscript (for details see adjustment and comments in the provided PDF).

Response: We agree with the reviewer and have addressed all the issues mentioned throughout the manuscript.

The authors shall reply to all points raised by the reviewer during 'Declined for Publication-Encourage Resubmission after revisions (Ref: toxins-812856 )' stage and 'Review round 1 (Ref: toxins-840002) next to the yellow and green color. Hereby, we are attaching the highlighted manuscript for resubmission and revision:

Reviewer 3 Report

The authors have greatly improved their manuscript.  It is now focused on the antimicrobial effects of bee venom and its components as antibacterial, antiviral and antifungal agents.

The tables are fairly comprehensive, but still suffer from not having acronyms defined.  MIC and MBC are examples of this.

The text still suffers from syntax errors.  Also, sometimes the range of number values are presented in high-low value format instead of low-high value format.  Please address.

The authors did not answer my question concerning their illustrations in the last paper.  So again, are the images presented original, not copyright protected, or parts of figures that are copyright protected? 

Author Response

Response to Reviewer 3 Comments:

The authors have greatly improved their manuscript.  It is now focused on the antimicrobial effects of bee venom and its components as antibacterial, antiviral and antifungal agents.

 Response: Thank you for your valuable comments

  1. The tables are fairly comprehensive, but still suffer from not having acronyms defined.  MIC and MBC are examples of this.

Response: We would like to thank the referee to provide such insightful guidance. We agree with the reviewer and have addressed this issue in a more appropriate way as follow:

  • Table 1:” MR “ changed to “Methicillin-resistant (MR)”, “ aeruginosa” to” Pseudomonas aeruginosa” , “A. baumannii ATCC 19606” to Acinetobacter baumannii ATCC 19606”
  • Lines 160-161: MIC: Minimum inhibitory concentration; MBC: minimum bactericidal concentrations; NR: No reported  
  • Table 2: Line 216-217: “EC50: Effective concentration for 50% reduction; ID50: 50% inhibitory dose; IC50: Inhibition concentration of 50%.”
  • Table 3: Line 242: “ppm: Parts per million”
  1. The text still suffers from syntax errors.  Also, sometimes the range of number values are presented in high-low value format instead of low-high value format.  Please address.

Response: Thanks so much for your kind evaluation. The concerns were addressed as:

  • Lines 141-144: “melittin, co-amoxiclav, ceftazidime, piperacillin, imipenem, ciprofloxacin and netilmicin are 4, 16, 32, 128, 16, 8 and 16 µg/ml, respectively.“ to “Melittin 's antibacterial and synergistic effects with ß-lactam antibiotics to baumannii was reported using microbroth dilution method. The MIC values of melittin, ciprofloxacin, co-amoxiclav, imipenem, netilmicin, ceftazidime, piperacillin are 4, 8, 16, 16, 16, 32, and 128 µg/ml, respectively”.
  • Lines 145:“antibacterial drugs are 0.312, 0.187, 0.375, 0.250, 0.750, and 1.25 µg/ml, respectively” to “antibacterial drugs are 0.750, 0.312, 0.250, 1.25, 0.187, and 0.375 µg/ml, respectively”.
  • Lines 193-194: “activity of Hepatitis C virus (HCV), Dengue virus (DENV) and Japanese encephalitis virus (JEV) with IC50 values of 117, 183 and 49 ng/ml, respectively” to “the activity of Japanese encephalitis virus (JEV), Hepatitis C virus (HCV), and Dengue virus (DENV) with IC50 values of 49, 117, and 183 ng/ml, respectively”
  • In lines 201-204: “In another in vitro study, melittin was evaluated toward different viruses: VSV-GFP, PR8-GFP, RSV-GFP, HSV-GFP, H3-GFP, and EV-71 with EC50 values of 1.18 ± 0.09, 1.15 ± 0.09, 0.35 ± 0.08, 0.94 ± 0.07, 0.99 ± 0.09, and 0.76 ± 0.03 μg/ml respectively,” change to “different viruses: Respiratory syncytial virus (RSV), Enterovirus-71 (EV-71), Herpes simplex virus (HSV), Coxsackie virus (H3), Fused Influenza A virus (PR8), and Vesicular stomatitis virus (VSV) with EC50 values of 0.35 ± 0.08, 0.76 ± 0.03, 0.94 ± 0.07, 0.99 ± 0.09, 1.15 ± 0.09, and 1.18 ± 0.09 μg/ml respectively”.
  1. The authors did not answer my question concerning their illustrations in the last paper.  So again, are the images presented original, not copyright protected, or parts of figures that are copyright protected? 

Response: The authors confirm that the figures are original.

Round 2

Reviewer 2 Report

The referee has received a poor point-to-point response for the second round of revision. Amongst other, the authors response letter does not fully reflect the entire text of the referees comments, which makes my job very hard.  The manuscript has improved slightly however, the referee noticed minor ambitions from the authors to really improve their work. Mainly, the authors removed many text and Figure 2 instead of critically discuss the topic and to implement the raised questions by the referee.

Find here some additional suggestions referring to the provided point-to-point response:

  •   lines 46-48: please rephrase the last sentence. The phrase bee products is not scientifically sound. 
  • lines 196-198: The Na,K-ATPase is a mammalian channel from the host cell, correct. 
  • lines 221-onward: The authors have not sufficiently commented on the referees questions. The manuscript points several times on possible application of BV or BV derived peptides in a therapeutic setting on human beings. Therefore, it is utmost important to critically consider the possible cytotox. and haemolytic activity of BV and its major metabolite melittin. 
  • Figure 2 was removed from the resubmitted version, but the referee suggested to provide structures, amino acid sequences of melittin, and other BV peptides....
  • lines 225-227: The ultra fast acting (5min) melittin uses which mode of action (haemolysis?). Maybe the authors have identified a hugh negative aspect of melittin applications compared to fluconazol.
  • lines 256-258: rephrase off-target side effects - this phrasing is not good. What cosmetic therapeutic applications? Is that phrase scientific sound?

Author Response

Response to Reviewer 2 Comments:

The referee has received a poor point-to-point response for the second round of revision. Amongst other, the authors response letter does not fully reflect the entire text of the referees comments, which makes my job very hard.  The manuscript has improved slightly however, the referee noticed minor ambitions from the authors to really improve their work. Mainly, the authors removed many text and Figure 2 instead of critically discuss the topic and to implement the raised questions by the referee.

Response: We would like to explain that we have tried to respond fully to the previous round of revision and comply with the reviewers` comments. We omitted some parts and one figure upon the request of the other reviewers. We would like to apologize if our actions did not meet your expectations, but we are ambitious to improve and refine the content and context of our work. So please do not hesitate to pay our attention to your concerns and questions.

NB: The manuscript has been edited by native English speakers

Find here some additional suggestions referring to the provided point-to-point response:

1- Lines 49-51: please rephrase the last sentence. The phrase bee products is not scientifically sound.

Response: the sentence was rephrased to “Natural products including beevenom (BV), one of many bee products which is rich of bioactive compounds, offer a diversity of activities against variety of diseases causes [3–6].” Lines (47-48)

2- Lines 196-198: The Na,K-ATPase is a mammalian channel from the host cell, correct. 

Response: “Na+, K+ pumps of the infected cells.” changed to “Na+, K+ pumps of the host cells” Line (196)

3-Lines 221-onward: The authors have not sufficiently commented on the referees questions. The manuscript points several times on possible application of BV or BV derived peptides in a therapeutic setting on human beings. Therefore, it is utmost important to critically consider the possible cytotox. and haemolytic activity of BV and its major metabolite melittin. 

Response: “Therefore, BV and melittin are attractive therapeutic candidate for microbial diseases. However, using BV and melittin induces extensive hemolysis and toxicity of the cells, a severe side effect that limited their future development and clinical application.” Lines (256-258)

4- Figure 2 was removed from the resubmitted version, but the referee suggested to provide structures, amino acid sequences of melittin, and other BV peptides....

Response: The chemical structure and amino acids sequence of melittin and apamin (Fig 1) Lines (154-156).

5- Lines 225-227: The ultra-fast acting (5min) melittin uses which mode of action (haemolysis?). Maybe the authors have identified a hugh negative aspect of melittin applications compared to fluconazol.

Response: Melittin safety in animal studies should be evaluated, and the efficacy of melittin in comparison to fluconazole is reported in Table 3.

Lines 227-228: The statement is describing bee venom and not melittin.

6- Lines 256-258: rephrase off-target side effects- this phrasing is not good. What cosmetic therapeutic applications? Is that phrase scientific sound?

Response: “BV and its constituents in combination with antibiotic drugs emerges as a plausible approach to overcome drug resistance of current antibiotics treatment in a controlled manner. Another promising and feasible implication is to test BV to combat microbes causing skin diseases. Interestingly, BV can be useful as a topical agent for encouraging skin regeneration or treatment of certain epidermal conditions [5,94]. Therefore, BV has been contributed to some formulations against  bacteria that causes acne [95,96].” Lines (250-255).

Reviewer 3 Report

No further comments.

Author Response

Response to Reviewer 3 Comments:

No further comments.

Response: we would like to thank the referee for the positive feedback.

This manuscript is a resubmission of an earlier submission. The following is a list of the peer review reports and author responses from that submission.

Round 1

Reviewer 1 Report

In this review the author(s) intend to describe a great number of medical (antimicrobial) applications of crude or individual components of bee’s venom. Thou the subject been reviewed brings some interesting information, the manuscript is poorly written and requires profound revision of its structure as well as English style.

  1. Several inappropriate words or terminology were used trough out the text, for instance: the word ingredients instead of compounds to indicate some individual venom’s components (lines 5 and 6 and several other places in the text); popular bacterial pathogens (line 20); encouraging skin recovery, and many others.
  2. Various confuse sentences and sometimes whole paragraph: line 28, the word comprises; lines 31-34; lines 40-57: several confusing sentences; lines 201-202: PLA2 as a peptide? Lines 239 -240: “and in another study, and its melittin showed antimicrobial activity…”; lines 244-245: “and induction the phosphorylation of Smad 2/3 and p38 MAPK; lines 280 and 402: starting sentences with numbers.
  3. Repeated information about the same subject: lines 93-95, 108-112, 144-146 and 151-156.
  4. Non sense data: line 97: numbers 1.024-256; line 191-192: “give inhibition 
of 0.35±0.06-fold”. 

  5. Chapter 4.1 Atopic Dermatitis (Lines 255 – 296): differently from other parts of the review manuscript, here the whole text was describing detailed results of experiments, giving the impression that it is actually the description of a regular article that should be submitted as such.
  6. Line 328: what means the number 80 here?
  7. Figure 1: what means challenges here?
  8. Lines 120 – 141: The great amount of written information could be summarized is tables.
  9. Line 328: Reference 99 is incomplete. It does not include the journal.

Reviewer 2 Report

The brief review on

The work entitled Antimicrobial properties of Apis mellifera`s bee  venom and their potential applications is presented to fill the absence of reviews bringing together the main articles on which address the effects of bee venom and its constituents as antimicrobials, their possible mechanisms of action and therapeutic applications.

The brief review has a wide approach, discussing, in especial, the antibacterial activity of BV and of its main components (melittin, apitoxin, and BV-PLA2), demonstrated by different authors in vitro experiments, and suggesting the use of BV in combination with antibiotics,  potentiating their effects. As a consequence, it may allow lowing the dose of the chemotherapy, highlighting the effect of BV on MRSA strains. The table showing the main results in this area is clear and illustrative.

The anti-viral activity , in vitro and in one in vivo experiment was also presented in this work. The discussion on the effects and mechanism of action of BV-PLA2, and its derivatives against HIV (in vitro) and  melittin to H1N1  (in vivo), and other viruses, are of potential interest for many readers. (Please see the comment bellow about Figure 3). Again, the Table helps to summarize quite well the revision on that point.

The authors also explore, more briefly, the works already published on the anti-fungal activity of BV, and Table 3 resumes the checked data.

The last topic of this review was dedicated to the therapeutic use of BV and fractions in dermatologic-related disorders that the authors entitled “Cosmetic applications” and, although for some points the use was used for cosmetic purposes, most of the examples are of actually skin diseases. I suggest changing the sub-title of this topic. The topic addresses several studies in which the use of BV and melittin was done in different disorders like atopic dermatitis, acne vulgaris, vitiligo and psoriasis, alopecia, wound healing, and also in cosmetic applications, like facial wrinkles. Worthy to mention that the text on atopic dermatitis contains more details on the data collected, being more extensive and non-proportional in comparison with the other subtopics.

Overall, the review work is updated and offers to readers enough information.

However, some minor points should be addressed to improve the text and reading.

- Figure 1 is out of place. This figure should be at the end of the last topic of the review (Cosmetic application), since it refers to the data discussed there.

- Consider reviewing references 30 and 31, which are not the most appropriate to cite the anti-inflammatory effect of BV.

- It is necessary to revise the paragraph between lines 151 and 165, where there are several truncated sentences,

- There is no need for Figure 3. The points highlighted in the figure could be better addressed in the text itself.

- Page 11, line 243: ECM (extracellular matrix?) Is not a chemical mediator.

Reviewer 3 Report

The referees comments are enclosed in the provided document. In general, this review article needs substantial reorganization. The authors should more clearly work out the main purpose of the work. The figures needs to be improved in quality and in the transferred informations. the text needs proper language check and gammer of many sentences is wrong. Please, consider to hire an expert for this work. The review is not critical at all but in this current version many informations are just provided without comments from the authors. 

The reader will be interested in:

  • the mode of action of BV and BV derived peptides or enzymes
  • the current state of the art regarding in vitro bioactivity characterization
  • the current state of the art in in vivo studies as well as in clinical investigations
  • an outlook on upcoming BV research 

Reviewer 4 Report

Major:

  1. More than half of the paper is dedicated to the cosmetic applications of the bee venom. According to the title, it does not fall into the scope of this manuscript. How does cosmetics applications of the bee venom is related to antimicrobial properties of the bee venom, such as wrinkles and alopecia?
  2. The manuscript contains typos, grammar and speech mistakes. Please, revise it.
  3. Line 49. Please clarify “BV and its constituents have a synergetic effect with chemotherapy agents due to their antibacterial properties”
  4. Line 54. “…represent early weapons against various diseases like cancer, inflammatory, microbes, etc.” Please, change to “microbe invasion”.
  5. Pictures used in Figure 1 and 2 are clearly taken from the internet. This is unacceptable. Have you obtained the authors’ agreement to use each of the pictures?
  6. Describe what “MIC” stands for, when you use it for the first time.
  7. Figure 2 – please be consistent with using verbs in the figure. Change “downregulation” to “downregulates”, “upregulation” to “upregulates” etc.
  8. All figures in this manuscript look very inconsistent, as if they are taken out of different papers.
  9. Figure 3. Please add s at the end of each verb.
  10. Figure 3. What do you refer as “immunodeficiency” in the last block? Is it HIV? Please, specify.
  11. Table 2. Why “bee venom” is stated as “isolated compound” from the bee venom in the first column? This is confusing.
  12. Table 2. Please, be consistent and add s at the end of each verb.
  13. Line 232. “Natural products including plants, marine, microorganism and bee products have been considered”. The word choice in this sentence is very confusing. Re-write the sentence.
  14. In the paragraph 4.1 too many unnecessary specifics for the experiments are given.
  15. Line 328. “disorder that affects more than 80 and 42% of 328 Caucasian men..”. What does number 80 refer to?
  16. Line 377. “The application of BV-containing cosmetics according to this study was safe, successful…”. Please, describe what means “successful” in that study.
  17. Line 418. “Clinical studies of BV are relatively slow”. Please, use a better word than “slow”.
  18. The purpose and the hidden message in the Figure 1 is very unclear. Re-do the figure.

Minor:

  1. The text contains many mistakes and typos. Line 38 – an and; B
  2. Line 158. Please, change “viral related diseases” to viral diseases.
  3. Line 184. “suggesting an immensurable source of anti- viral”. Noun is missing at the end of the sentence.
  4. Line 254. “illustrated below:” – there is no picture illustrated below.
  5. Line 389. “approximately2-fold increase” – add space.
  6. Line 415. Change to “plays”.
  7. Line 418. “BV's synergy activity widely in cancer and microbial diseases.” – this sentence requires a verb.

Reviewer 5 Report

The authors present a review of the potential medicinal uses of bee venom, with a particular focus on its antimicrobial properties.  The rationale for the investigation was that bee venom may offer an alternative to antibiotics and other agents that infectious microbes may be developing a resistance to.

In general, the manuscript states its goals reasonably well, and the text flows accordingly.  I am concerned if the illustrations are all original or are composed of images obtained from copyright protected sources and then collaged into a figure.

The title implies what appears to be a review of antimicrobial properties of bee venom.  However, the figures and text also discuss the anti-inflammatory and cosmetic used for venom as well.  The title should be modified accordingly to reflect the content of the article.

The prose of the article is wanting, and the proposal that bee venom will work in settings that include acutely and chronically lethal viral infections seems very optimistic.  Frankly the article seems to tout bee venom very much as a cure for dozens of disease entities and states that clinically are not likely to respond to such therapy.

I have a few minor comments as well.

Abstract, line 12.  The authors use the word “microbial” when perhaps “antimicrobial” is what was meant.

Line 58. “In this review, we will discuss the in-vitro, in-vivo and in-situ implications of BV.”  The authors should add to this sentence something like “…therapeutic implications of BV administration.”  They need to indicate what sort of implications we should be interested in.

Lines 108-110.  “Melittin has a greater sensitivity to the gram-positive compared to gram-negative microorganisms due to the nature of the organism's cell membrane and the involved peptides [34,52,54,55].”  The authors are occasionally saying the opposite of what they mean.  The bacterial is sensitive to melittin, not the other way around.

Line 120. “…and analogue…”  A typo – an, not and.  The work is riddled with such errors.  The authors need to correct such problems.

In sum, it would be better if the authors focused more on perhaps the antibacterial and limited antiviral properties of bee venom instead of this very wide-ranging review of many possible but not probable therapeutic roles this treatment may deliver.